# A Missense Mutation of the *HSPB7* Gene Associated with Heat Tolerance in Chinese Indicine Cattle

**DOI:** 10.3390/ani9080554

**Published:** 2019-08-14

**Authors:** Lulan Zeng, Yanhong Cao, Zhuyue Wu, Mingguang Huang, Guoliang Zhang, Chuzhao Lei, Yumin Zhao

**Affiliations:** 1Key Laboratory of Beef Cattle Genetics and Breeding in Ministry of Agriculture and Rural Agriculture, Branch of Animal Husbandry, Jilin Academy of Agricultural Sciences, Changchun 130033, China; 2College of Animal Science and Technology, Northwest A&F University, Yangling, Shaanxi 712100, China; 3The Animal Husbandry Research Institute of Guangxi Zhuang Autonomous Region, Nanning 53001, China

**Keywords:** Chinese cattle, *HSPB7* gene, heat tolerance

## Abstract

**Simple Summary:**

A missense mutation (NC_037329.1: g.136054902 C > G: p. Ala69Gly) was identified in the heat shock protein family B (small) member 7 (*HSPB7*) gene in indicine cattle, which might be a candidate mutation associated with the heat tolerance. Here, Polymerase Chain Reaction and DNA sequencing methods were used to detect this mutation in 774 individuals belonging to 32 Chinese indigenous cattle breeds. The distribution of alleles of NC_037329.1: g.136054902 C > G displays significant geographical difference across native Chinese cattle breeds and cattle carrying allele G distributed in regions with higher mean annual temperature, relative humidity, and temperature humidity index. Our results demonstrate that the mutation of the *HSPB7* gene in Chinese indicine cattle might be a candidate gene associated with the heat tolerance.

**Abstract:**

The small heat shock proteins (HSPB) are expressed in response to heat stress, and the heat shock protein family B (small) member 7 (*HSPB7*) gene has been reported to play an important role in heat tolerance pathways. Only a missense mutation (NC_037329.1: g.136054902 C > G: p.Ala69Gly) was identified in the *HSPB7* gene in indicine cattle, which might be a candidate mutation associated with the heat tolerance. Here, we explore the allele frequency of this mutation in 774 individuals belonging to 32 Chinese indigenous cattle breeds using polymerase chain reaction (PCR) and DNA sequencing methods. The distribution of alleles of NC_037329.1: g.136054902 C > G displays significant geographical difference across native Chinese cattle breeds that the allele C was dominant in northern cattle groups, while allele G was dominant in southern indicine cattle groups. Additionally, the association analysis indicated that the G allele was significantly associated with mean annual temperature (T), relative humidity (RH), and temperature humidity index (THI) (*p* < 0.01), suggesting that cattle carrying allele G were distributed in regions with higher T, RH, and THI. Our results demonstrate that the mutation of the *HSPB7* gene in Chinese indicine cattle might be a candidate gene associated with the heat tolerance.

## 1. Introduction

In tropical and subtropical areas, such as southern China, one of the problems encountering the livestock population is the excessive ambient temperature and humidity during summer. Negative effect caused by heat stress can compromise a variety of physiological functions including milk yield [1], reproduction [2], and immune function [3], causing a tremendous economic loss. The heat tolerance in animals is a quantitative trait [4,5,6], so genetic factors are important to the phenotype of heat tolerance.

All organisms express large numbers of heat shock proteins (HSPs) and respond to heat shock during acute and persistent stimulation by high temperatures. The small heat shock proteins (HSPB) belong to a highly conserved protein family that is expressed throughout embryonic development and in adult organs. It plays an important role in interacting with components of the cytoskeleton [7,8,9,10,11] and maintains protein homeostasis within the cell [12,13]. The heat shock protein family B (small) member 7 (*HSPB7*) is expressed exclusively in the heart [14,15,16]. As demonstrated previously, heat stress is associated with a tremendous increase in cardiac workload [17]. Thus, it is possible that the *HSPB7* gene expressed in the heart may have a role in heat tolerance [18]. To date, only a missense mutation c.206C > G (NC_037329.1:g.136054902 C > G, p.Ala69Gly) was identified in the coding region of the bovine *HSPB7* gene in indicine cattle by whole genome resequencing (http://animal.nwsuaf.edu.cn/code/index.php/BosVar). China possesses abundant genetic resources of cattle breeds, including 53 identified indigenous cattle breeds [19]. These cattle breeds distribute in vast areas, covering several temperature zones with notable temperature and humidity differences. Therefore, it is very suitable to detect functional SNP of the bovine *HSPB7* gene and determine its possible association with heat tolerance in Chinese cattle.

To date, the published information of the *HSPB7* gene in Chinese native yellow cattle remains limited. The aim of the present study was to explore the allele distribution of the missense mutation (p.Ala69Gly) of the bovine *HSPB7* gene in native Chinese cattle and investigate the association between different genotypes and mean annual temperature (T), relative humidity (RH), and temperature humidity index (THI), which will possibly contribute to evaluating it as genetic marker in heat tolerance for cattle breeding and genetics. 

## 2. Materials and Methods

### 2.1. Ethics Statement

The protocols used in this study and for the animals were recognized by the Faculty Animal Policy and Welfare Committee of Northwest A&F University (FAPWC-NWAFU, Protocol number, WAFAC1008).

### 2.2. Data Collection, DNA Extraction, and PCR Reaction

In all, 774 ear tissues representing 34 different cattle breeds were collected from preservation regions and state-owned farms (Appendix A). Genomic DNA was extracted using phenol-chloroform method as described by Sambrook and Russel [20]. According to the published nucleotide sequence information of the bovine *HSPB7* gene (GenBank no. NC_037329.1), a pair of oligonucleotide primer was synthesized and the detail information of oligonucleotide sequence, annealing temperature, and fragment size was depicted in Appendix A. The PCR reaction was performed in a 25 μL volume containing 20 ng of genomic DNA, 20 pM of each primer, 0.2 mM of deoxy-ribonucleotide triphosphates (dNTPs) (Takara, Dalian, China), 1 × polymerase chain reaction (PCR) buffer (including 2.5 mM of Mg^2+^) (Takara, Dalian, China), and 1.0 U of rTaq DNA polymerase (Takara, Dalian, China). The cycling protocol was 5 min at 95 °C, following by 30 cycles of 94 °C for 30 s, annealing at 60 °C for 30 s, and 72 °C for 30 s, with a final extension at 72 °C for 10 min. The PCR products were detected by electrophoresis on a 1.5% agarose gel including 0.5 μg/mL of ethidium bromide (Takara, Dalian, China), visualized under UV and photographed. Then, PCR productions were directly sequenced with the ABI PRIZM 377 DNA sequencer (Perkin-Elmer, Shanghai Sangon Biotech Company, Shanghai, China). 

### 2.3. Statistical Analysis

Based on nucleotide analysis of 32 indigenous cattle breeds, genotype and allele frequencies were determined. THI is a useful and easy measure to ascertain heat load intensity of thermal climatic conditions by evaluating the combined effects of T and RH. Then, it was calculated according to the formula developed by National Oceanic and Atmospheric Administration [21]:THI = (1.8T + 32) − (0.55 − 0.0055 RH) (1.8T − 26)
where T is temperature in degrees Celsius and RH is relative humidity as a percentage.

T and RH over the last 30 years of sampling site of 32 cattle breeds were collected from the Chinese Central Meteorological Office (Appendix A) (http://data.cma.cn/) [22].

Environmental association analyses were performed for three climatic variables with 3 genotypes [23,24]; we performed the general linear model (GLM) implemented in the SPSS 18.0 software (SPSS, Inc, Armonk, US). As described in Eckert et al. 2010 [25], we treated the environment variable as a phenotype. The statistical model is:Climatic variable = Marker effect + Residual
where Climatic variable are the values of T, RH, and THI between 1951 and 1980; Marker effect = the fixed effect of the genotypes; Residual is the random residual effect. Differences were considered significant at *p* < 0.01.

## 3. Result

In our study, a novel mutation (NC_037329.1: g.136054902 C > G) was validated by DNA sequencing in the *HSPB7* gene among Chinese cattle, causing amino acid substitution Ala to Gly. The genotype and allele frequencies of NC_037329.1: g.136054902 C > G are given in Appendix A. At NC_037329.1: g.136054902 C > G locus, three genotypes (CC, GC, and GG) were detected and the mean allele frequencies for C and G in Chinese cattle were 0.5807 and 0.4193, respectively. As compared to taurine cattle, indicine cattle are more thermo-tolerant for their superior ability in regulating body temperature in response to heat stress [26]. In this study, pure taurine (Angus) and indicine cattle (Zebu) were used to determine the species-specific allele frequencies. This nonsynonymous mutation, in contrast to the pattern observed in Angus breed, has nearly reached fixation (96.25%) in the Zebu populations (Figure 1), which showed that the GG genotype nearly fixed in Zebu populations.

The relationship between SNPs and three environmental parameters (T, RH, and THI) were analyzed (Table 1). There was a significant (*p* < 0.01) association between three environmental parameters and allele G. The results suggested that individuals with GG or GC genotype distributed in the area with a significantly higher value of T, RH, and THI compared with those with genotype CC (*p* < 0.01), revealing the G allele might be related to heat tolerance in Chinese indigenous cattle. Test of three subjects’ effects on *HSPB7* genotypes showed that the mean annual humidity has the strongest correlation with the genotypes (Appendix A). 

## 4. Discussion

Due to climate change and an ageing population, heat exhaustion or heatstroke caused by excessive heat or physical exertion in individuals is becoming more apparent in recent years. Sudden death related to heat stroke occurs primarily in individuals with pre-existing cardiac abnormalities, organ failure secondary to rhabdomyolysis, or acute onset of organ failure [27]. For this reason, adverse cardiac events occur at a higher frequency during summer [28,29], and patients with cardiovascular disease are particularly vulnerable to injury from heat stress [30]. Three separate genome-wide association studies also have implicated that polymorphisms in the *HSPB7* gene may lead to idiopathic cardio myopathies [14,31,32]. Therefore, the *HSPB7* gene is very likely to be a candidate gene for heat tolerance in cattle. 

Northern China possesses cold and dry climate, whereas southern China is characterized as humid subtropical with periods of high ambient temperature and relative humidity [22]. Based on geographical distribution and morphological characteristics, Chinese cattle breeds also have been divided into northern, central, and southern groups [30]. Previous studies have confirmed that northern Chinese cattle are of the taurine whereas southern Chinese cattle belong to the indicine cattle [19]. The two populations expanded and hybridized, forming a hybrid zone in central China [33,34,35,36,37]. In our study, the allele frequencies (G) of northern, central, and southern groups were 0.0237, 0.2266, and 0.6143, respectively, which gradually diminish in Chinese indigenous cattle from south to north and exhibit a clear north–south gradient geographic structure across China.

## 5. Conclusions

Our results showed that the mutation of the *HSPB7* gene was associated with heat tolerance in Chinese cattle and could be a potential useful genetic marker to improve heat tolerance trait in cattle breeding programs.

## Figures and Tables

**Figure 1 animals-09-00554-f001:**
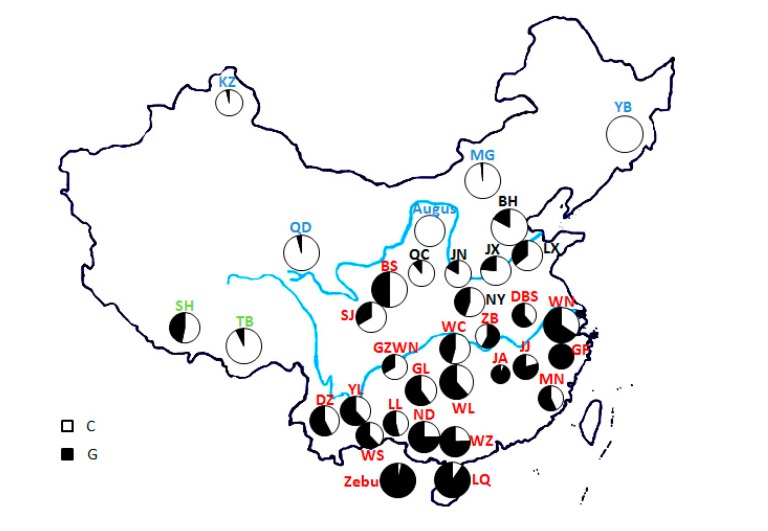
Geographical distribution of NC_037329.1:g.136054902 C > G among 32 Chinese breeds as well as Angus and Indian Zebu population. Note: KZ = Kazakh; YB = Yanbian; QD = Qaidam; MG = Mongolian; JX = Jianxian red; NY = Nanyang; LX = Luxi; BH = Bohai Black; JN = Jinnan; QC = Qinchuan; JA = Ji’an; JJ = Jinjiang; WN = Wannan; GZWN = Weining; ZB = Zaobei; DBS = Dabeishan; BS = Bashan; LL = Longling; WZ = Weizhou; ND = Nandan; YL = Yunling; LQ = Leiqiong; WS = Wengshan; DZ = Dianzhong; GF = Guangfeng; SJ = Sanjiang; WL = Wuling; GL = Guanling; WC = Wuchuan; MN = Minnan; TB = Tibetan; SH = Shigatse Humped.

**Table 1 animals-09-00554-t001:** Association of genotypes of the *HSPB7* NC_037329.1: g.136054902 C > G locus with mean annual temperature (T), humidity (H), and temperature–humidity index (THI) in Chinese indigenous cattle.

Polymorphism	Genotypes (Number)	Temperature (°C) (LSM ± SE)	Relative Humidity (%) (LSM ± SE)	Temperature–Humidity Index (LSM ± SE)
*HSPB7*	CC (328)	9.980 ^C^ ± 0.267	62.332 ^C^ ± 0.585	51.961 ^C^ ± 0.369
NC_037329.1:	CG (243)	14.870 ^B^ ± 0.310	72.181 ^B^ ± 0.680	58.367 ^B^ ± 0.428
g.136054902 C > G	GG (203)	17.367 ^A^ ± 0.339	76.744 ^A^ ± 0.744	62.072 ^A^ ± 0.469

LSM ± SE represent the least square means with standard errors for diverse genotypes and environment parameters. Means in the same column and locus with difference capital superscripts, A, B, and C, are different at *p* < 0.01.

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
