# Peer review of "A Missense Mutation of the HSPB7 Gene Associated with Heat Tolerance in Chinese Indicine Cattle"

_animals, 2019, doi:10.3390/ani9080554_

Round 1

Reviewer 1 Report

In the current manuscript, the authors survey the frequency and geographical distribution of a misses mutation in a heat shock protein HSPB7. The motivation of this survey is to identify candidate genes associated with heat tolerance, which might in future be useful for animal breeding. The authors have run a statistical model but have not explained what are the terms and parameters returned from their model signify biologically. The introduction/results sections can be used to explain them better. More detail ed comments below: The key results depend on the model in line 91, and the results in Table 1. The way the current model is set, up, Y is the dependent variable (y-axis variable) while the genotype G is the independent variable (x-axis variable). Should not the model be the other way around, as it is the Temperature that is causing the Genotype? Hence genotype should be the dependent variable, and therefore on the left side of the equation? The strength of association might still end up being numerically the same but wrong in principle as per authors’ model. In table 1, it is not clearly specified what is the value for which the LSM is being calculated? Is it the “strength of association between T/H/THI and genotype”? In what units is it being measured? What is the difference that is being compared? Is it the LSM of each genotype e.g. CC against other genotype e.g. GG ? What do the superscript letters A, B, C in Table 1 signify ? It should be explained in the legend. Is it possible for authors to make additional figures like Figure 1, showing trends in each of the variables T, H and THI across China? It will make the visualization of correlation between genotype distributions and T,H, THI more clear. However, this is not essential for the authors to do. Minor comments: 1. Line 53, 54 : Please replace “Chen et al. unpublished” and “Chen et al.” with reference 19 or as appropriate. 2. Line 79: Provide full expansion of the abbreviation THI when it is mentioned for the first time. 3. Figure 1 label on the map and in the legend: Please correct “Augus” to “Angus”

Author Response

The key results depend on the model in line 91, and the results in Table 1.

The authors have run a statistical model but have not explained what are the terms and parameters returned from their model signify biologically.

The way the current model is set, up, Y is the dependent variable (y-axis variable) while the genotype G is the independent variable (x-axis variable). Should not the model be the other way around, as it is the Temperature that is causing the Genotype? Hence genotype should be the dependent variable, and therefore on the left side of the equation? The strength of association might still end up being numerically the same but wrong in principle as per authors’ model.

Reply: Thank you for underlining this deficiency.

In our study, we performed environmental association analyses for three climatic variables with 3 genotypes; mean annual temperature (T), relative humidity (RH) and temperature humidity index (THI). We performed the general linear model (GLM) implemented in SPSS. As in (Eckert et al. 2010; Hancock et al. 2011; Westengen et al. 2012), we treated the environment variable as a phenotype and the statistical model is:

Climatic variable = Marker effect + residual
where Climatic variable are the values of T, RH, and THI between 1951 and 1980; Marker effect = the fixed effect of the genotypes; residual is the random residual effect.

Eckert A.J., Joost V.H., Wegrzyn J.L., C Dana N., Jeffrey R.I., González-Martínez S.C. & Neale D.B. (2010) Patterns of population structure and environmental associations to aridity across the range of loblolly pine (Pinus taeda L., Pinaceae). Genetics 185, 969.

Hancock A.M., Benjamin B., Nathalie F., Horton M.W., Jarymowycz L.B., F Gianluca S., Chris T., Fabrice R. & Joy B. (2011) Adaptation to climate across the Arabidopsis thaliana genome. Science 334, 83-6.

Westengen O.T., Berg P.R., Kent M.P. & Brysting A.K. (2012) Spatial structure and climatic adaptation in African maize revealed by surveying SNP diversity in relation to global breeding and landrace panels. PloS one 7, e47832.

In table 1, it is not clearly specified what is the value for which the LSM is being calculated? Is it the “strength of association between T/H/THI and genotype”? In what units is it being measured?

What is the difference that is being compared? Is it the LSM of each genotype e.g. CC against other genotype e.g. GG? What do the superscript letters A, B, C in Table 1 signify? It should be explained in the legend.

Reply: Thank you for your suggestions. In table 1, LSM ± SE represent the least square means with standard errors for diverse genotypes and environment parameters. Means in the same column and locus with difference capital superscripts, A, B and C, are different at P < 0.01. We have added it to this manuscript.

Is it possible for authors to make additional figures like Figure 1, showing trends in each of the variables T, H and THI across China? It will make the visualization of correlation between genotype distributions and T, H, THI clearer. However, this is not essential for the authors to do.

Reply: Thanks. Geographical distributions of three environmental parameters (T, H and THI) over the last 30 years in China were drew in previous study (Zeng, L., et al., PRLH and SOD 1 gene variations associated with heat tolerance in Chinese cattle. Animal Genetics, 2018. 49(5): p. 447-451) which was been cited in our manuscript (Line).

Minor comments:

Line 53, 54 : Please replace “Chen et al. unpublished” and “Chen et al.” with reference 19 or as appropriate.

Reply: Thank you for underlining this deficiency. we corrected the mistake

Line 79: Provide full expansion of the abbreviation THI when it is mentioned for the first time.

Reply: Thank you for underlining this deficiency. we corrected the mistake.

Figure 1 label on the map and in the legend: Please correct “Augus” to “Angus”

Reply: Thanks, we have corrected this mistake.

Reviewer 2 Report

The short communication entitled “A missense mutation of HSPB7 gene associated with heat tolerance in Chinese indicine cattle” by Lu-lan Zeng has been well-prepared. Therefore, there is no specific comments from this reviewer.

To my knowledge, there is no paper describing the expression pattern of HSPB7 in cattle tissues. If possible, the authors can include the PCR based expression of HSPB7 in different samples (including heart) of representative taurine and indicine cattle.  

Line 103. specie-specific allele frequency. Please correct the typo.

Author Response

Reviewer 2:

To my knowledge, there is no paper describing the expression pattern of HSPB7 in cattle tissues. If possible, the authors can include the PCR based expression of HSPB7 in different samples (including heart) of representative taurine and indicine cattle.

Reply: Thanks, it is a good idea. However, different samples of cattle tissues were not collected before and Chinese cattle distribute in vast area. It is a huge work to collect so many samples within few days, so we cannot add this experiment in this study.

Line 103. specie-specific allele frequency. Please correct the typo.

Reply: Thanks. we have corrected the mistake.